# Purification and Characterisation of Badger IgA and Its Detection in the Context of Tuberculosis

**DOI:** 10.3390/vetsci6040089

**Published:** 2019-11-02

**Authors:** Deanna Dalley, Sandrine Lesellier, Francisco J. Salguero, Mark A. Chambers

**Affiliations:** 1Department of Bacteriology, Animal and Plant Health Agency, Addlestone, Surrey KT15 3NB, UK; deanna.dalley@apha.gov.uk; 2Laboratoire de la Rage et de la Faune Sauvage de Nancy (LRFSN), ANSES, 54220 Malzéville, France; sandrine.lesellier@anses.fr; 3Public Health England, Porton Down, Salisbury SP4 0JG, UK; Javier.Salguero@phe.gov.uk; 4School of Veterinary Medicine, Faculty of Health and Medical Sciences, University of Surrey, Guildford GU2 7AL, UK

**Keywords:** immunoglobulin A, badger, tuberculosis

## Abstract

European badgers are a wildlife reservoir of bovine tuberculosis in parts of Great Britain. Accurate diagnosis of tuberculosis in badgers is important for the development of strategies for the control of the disease. Sensitive serological tests for badger TB are needed for reasons such as cost and simplicity. Assay of mucosal IgA could be useful for diagnosing respiratory pathogens such as *Mycobacterium bovis* and for monitoring the response to mucosal vaccination. To develop an IgA assay, we purified secretory IgA from badger bile, identifying secretory component (SC), heavy chain (HC) and light chain (LC), at 66, 46 and 27 Kda, respectively, on the basis of size comparison with other species. Monoclonal antibodies (mAbs) were generated to purified IgA. We selected two for ELISA development. The detection limit of the IgA-specific mAbs was found to be approximately 20 ng/mL when titrated against purified badger bile. One monoclonal antibody specific for badger IgA was used to detect IgA in serum and tracheal aspirate with specificity to an immunodominant antigen of *M. bovis*. An *M. bovis* infection dose-dependent IgA response was observed in experimentally infected badgers. IgA was also detected by immunohistochemistry in the lungs of bTB-infected badgers. With further characterisation, these represent new reagents for the study of the IgA response in badgers.

## 1. Introduction

In parts of Great Britain and Ireland, the European badger (*Meles meles*) constitutes a reservoir of infection for *Mycobacterium bovis* and a potential source of infection to cattle [1,2,3]. Accurate diagnosis of *M. bovis* infection in badgers is an important component of strategies to control bTB in this species. Culture isolation of *M. bovis* remains the “gold-standard” diagnostic test but this is only sensitive when post-mortem tissue samples are used. Thus, sensitive in vitro diagnostics that can be used to test live animals are still required.

Assays based on measurement of a cellular immune response are commonly used for the diagnosis of TB in cattle, humans and other mammals. In badgers, we have developed an interferon-gamma (IFNγ) release assay (IGRA) for bTB detection in badgers [4]. With a sensitivity of up to 81% and a specificity of 94% [5,6], it is the most accurate bTB test that can be performed on live badgers. However, as an alternative diagnostic approach, assays measuring serological responses offer several advantages. These include test rapidity and ease of use and the stability of antibodies during sample transport, processing and storage. Antibody-based assays for badger TB have been developed previously [7,8] but historically have lacked sensitivity (reviewed in [9,10]). Although more recent developments show promising enhancement in test accuracy [11], there still remains a need for more sensitive serological-based diagnostic tests for badger TB.

Most serological assays developed to date for badgers have evaluated the immunoglobulin G (IgG)-mediated humoral response against mycobacterial antigens. For example, the badger bTB ELISA measures IgG recognition of MPB83, a glycosylated lipoprotein that is a major target of the antibody response in *M. bovis* infected badgers [12], later confirmed using a multi-antigen print immunoassay (MAPIA) [13]. However, there is justification for the evaluation of the IgA response to mycobacterial infection as the basis of an improved serodiagnostic test for badger TB. When Conde et al. evaluated the usefulness of detection of serum IgA and serum IgG antibodies directed against the mycobacterial P-90 antigen for the diagnosis of pulmonary TB in people, they found that an IgA-based ELISA was more sensitive and specific than one based on IgG [14]. IgA is the predominant Ig isotype in human mucosal tissue and comprises about 60% of the total immunoglobulin produced in humans [15]. Since *M. bovis* is primarily a respiratory pathogen in badgers [16], it would be beneficial to develop an immunological test to detect the local response to infection in badgers. Furthermore, immunoglobulins, including IgA have been localised by immunohistochemical staining in bovine granulomatous lesions caused by *M. bovis* [17].

Studies in rats have revealed bile to be an abundant source of both secretory IgA (sIgA) and free secretory component (SC) [18]. For this reason, together with its accessibility and abundance, badger bile was chosen as the source of sIgA for the development of the ELISA for badgers. In this study, a panel of monoclonal antibodies (mAbs) was raised against badger sIgA purified from bile. The mAbs were then screened using purified sIgA and IgG in order to identify suitable mAbs for ELISA development. One mAb with reactivity to sIgA was selected to detect IgA in serum and tracheal aspirates by ELISA with specific recognition of recombinant antigen MPB83. This mAb was also peroxidase labelled and used to detect IgA by immunohistochemistry within *M. bovis* induced granulomas in lungs from badgers.

## 2. Materials and Methods

### 2.1. Purification of sIgA from Badger Bile

Bile was obtained from a bTB-free badger post mortem. Badger bile was clarified by centrifugation and then concentrated five-fold using a stirred cell Amicon concentrator (Sigma-Aldrich Company Ltd., Dorset, UK) without precipitation of the proteins. The concentrated preparation was dialysed three times against 2% NaCl, buffered with 0.02 M Tris-HCl pH 8.0 containing 0.1% Kathon. The dialysed concentrate was subjected to further centrifugation before application to a BioSep Sec 3000 HPLC gel filtration column (Phenomenex, Macclesfield, UK), equilibrated with 15 mM NaH_2_PO_4_, 45 mM Na_2_HPO_4_ and 0.15 M NaCl. Fractions were collected according to their absorbance at 280 nm. Those corresponding to the three main peaks detected by the UV-monitor were run on SDS-PAGE gels, silver-stained then transferred to nitrocellulose membranes for Western blotting. The badger bile IgA blots were probed with a panel of commercial anti-IgA and anti-SC polyclonal antibodies with specificity to dog or pig IgA (Bethyl Laboratories Inc., Cambridge, UK); to human IgA (Accurate Chemical and Scientific Co., city, NY, USA); or to cat, goat, and pan-species SC (Accurate Chemical and Scientific Co.) to establish any cross-reactivity to badger IgA.

### 2.2. Generation of Monoclonal Antibodies to Badger IgA

An HPLC purified pool of protein representing the light chain (LC) and SC of badger sIgA was used to immunise six BALB/c mice by the subcutaneous route. The mice were then boosted on four occasions, after 3, 5, 8 and 11 weeks. All mice were bled from the tail vein one week after the second and third boosts and the serum tested for the presence of purified badger sIgA by direct ELISA. One mouse was chosen based on its seroreactivity to sIgA, and given a final boost with the protein. Its spleen was removed four days later for the production of B-cell hybridomas. Fifteen stable B-cell clones were produced from the spleen of this mouse. All immunoglobulins produced by the clones were identified as IgG, subclasses IgG_1_ and IgG_2_ using an IsoStrip Mouse Monoclonal Antibody Isotyping Kit (Roche Applied Science, Penzberg, Germany).

### 2.3. Specificity of Hybridomas for Badger IgA and IgG in a Direct ELISA

Badger IgG was isolated from the serum of a badger that had tested positive against MPB83 by Brock Test ELISA [7] and purified using a Protein G Sepharose column (Abcam, Cambridge, UK). Purified badger IgG and sIgA, obtained from badger bile as described previously, were coated onto Maxisorp plates (Nunc, Roskilde, Denmark), 100 μL/well, diluted in 50 mM carbonate buffer pH 9.6 (1:1500 and 1:45, respectively), overnight at 23 °C. The coating included wells containing buffer alone, to which all other reagents were added to provide a baseline value for each antibody. After three washes with PBS (pH7.2) containing 0.05% Tween 20 (PBSTw20) plates were blocked with 200 μL 3% casein. After one hour at 37 °C, plates were washed three times with PBSTw20. Fifteen IgA hybridoma supernatants and one peroxidase conjugated anti-IgG monoclonal (CF2, APHA Weybridge, Addlestone, UK) were added in each of the IgG and IgA coated wells at a concentration of 100 μL/mL (1 in 100 dilution). Following incubation for an hour at 37 °C, plates were washed three times with PBSTw20 before addition of a commercial biotinylated anti-mouse IgG antibody (GE Healthcare Life Sciences, Little Chalfont, UK) diluted 1:2000 in PBSTw20, 0.5% BSA. After one hour at 37 °C plates were washed three times with PBSTw20 before addition of 100 μL/well of Streptavidin Horseradish Peroxidase (diluted 1:4000, GE Healthcare Life Sciences). The wells containing CF2 were incubated with PBSTw20, 0.5% BSA only. After one hour at 37 °C, plates were washed three times with PBSTw20, before the addition of 100 μL/well of tetramethylbenzidine (TMB) substrate (Fisher Scientific UK Ltd., Loughborough, UK). After ten minutes at 24 °C, the reaction was stopped by the addition of 100 μL 2 M H_2_SO_4_ per well. The OD value for each hybridoma was divided by the corresponding baseline value without IgA to obtain a ratio. In order to standardise reactions obtained between the hybridomas (with varying concentrations of antibody in the supernatant) each hybridoma was titrated from 1:100 to 1:3500 dilution alongside CF2 at 1/100 as previously. A dilution was selected for each supernatant that gave the same OD as CF2, at its standard working dilution in the Brock Test ELISA [7].

### 2.4. Source of Samples

Samples of blood serum, tracheal aspirate and lymph node (LN) tissue were obtained from experiments published previously involving the endobronchial challenge of captive badgers with *M. bovis*. In the first study, three groups of badgers were infected with different concentrations of *M. bovis* suspension (< 10 colony forming units (cfu), approximately 100 cfu, or approximately 3000 cfu), plus a fourth, unchallenged control group [19]. In the second study, three groups of badgers underwent challenge with *M. bovis* (range, 2600–4800 cfu): two groups vaccinated intramuscularly with BCG Danish strain 1331 at two different doses (2–8 × 10^5^ cfu or 10-fold higher) prior to challenge, and one unvaccinated control group [20]. Each study was approved by the local animal ethics committee. The first study (BROC1) was approved by the University College Dublin animal ethics committee prior to the study commencing in 2002, and the second study (VES2) was approved by the APHA animal ethics committee, prior to the study commencing in 2007. 

### 2.5. Detection of MPB83 Specific-IgA in Serum and Tracheal Aspirate by ELISA

Recombinant MPB83 (Lionex GmbH, Braunschweig, Germany) was coated onto maxisorb plates at a concentration of 0.5 µg/mL. After blocking, serum or tracheal aspirate samples from experimentally infected badgers were diluted in PBSTw20, 0.5% BSA at a concentration of 1:1 and 100 μL added to the wells for one hour. Two wells did not receive the samples, as negative controls. The plates were then washed PBSTw20 and the plates were then incubated with mAb (1/1), diluted 1:500. As previously, commercial biotinylated anti-mouse IgG antibody diluted 1:2000 in PBSTw20, 0.5% BSA was added. After one hour at 37 °C plates were washed three times with PBSTw20 before addition of 100 μL/well of streptavidin horseradish peroxidase. After one hour at 37 °C, plates were washed three times with PBSTw20, before the addition of 100 μL/well of tetramethylbenzidine (TMB) substrate (Fisher Scientific UK Ltd., Loughborough, UK). After ten minutes at 24 °C, the reaction was stopped by the addition of 100 μL 2M H_2_SO_4_ per well.

### 2.6. Immunohistochemistry Staining

Badger tissues were removed at necropsy and fixed in zinc salts fixative as previously described [21]. Fixed samples were embedded into paraffin wax and sectioned (4 µm thickness) onto positive charged microscope slides, de-paraffinised with xylene (VWR International, Lutterworth, UK) and rehydrated through graded ethanols (Hayman, London, UK) before endogenous peroxidase activity was blocked using a hydrogen peroxide/methanol solution (hydrogen peroxide 3% *v*/*w*, SLS, Nottingham, UK in methanol, VWR International) for 15 min.

For immunohistochemistry, the slides were assembled into Shandon coverplates (Shandon Scientific, Runcorn, UK) and washed with 0.85% Tris buffered saline pH7.6 with 0.05% Tween 20 (TBST) (VWR International). All steps were undertaken at room temperature. The sections were then incubated in normal goat serum (Vector laboratories, Peterborough, UK) diluted (1:66) in TBST for 20 min, before application of IgA monoclonal, pre-conjugated with peroxidase (1/1, APHA). The tissue sections were washed twice (5 min each) with buffer after each incubation. Immunolabelling was “visualised” by applying 3,3′ diaminobenzidine tetrahydrochloride (DAB), (Sigma-Aldrich Company Ltd., Dorset, UK) and 0.01% hydrogen peroxide (SLS) for 10 min. Slides were washed with purified water and counterstained with Mayer’s haematoxylin (Surgipath, Peterborough, UK) before being dehydrated through graded alcohols, cleared in Xylene and permanently mounted using DPX. Control sections with an irrelevant peroxidase conjugated antibody were used in each IHC run.

## 3. Results

### 3.1. Purification and Western Blot, of Badger sIgA

Dialysed concentrate of badger bile was applied to a BioSep Sec 3000 HPLC gel filtration column. Three main peaks were detected by the UV-monitor and the fractions collected and run on SDS-PAGE gels and silver-stained. The three fractions corresponded to the heavy chain (HC), the LC, and SC, based on comparison with the sizes of these molecules in other selected species for which data could found (Table 1). Their identity was confirmed using Western blots probed with a panel of commercial anti-IgA and -SC antibodies, of which only rabbit anti-dog identified HC and LC (Bethyl Laboratories Inc., Cambridge, UK) (Figure 1). Badger sIgA was comprised of heavy and light chain components with SC the most abundant protein associated with sIgA. Size comparison with published data from selected other species showed badger HC, LC and SC to be most similar to baboon (Table 1).

### 3.2. Recognition by IgA Monoclonal Antibodies of Purified IgA and IgG

The supernatants from each of the fifteen B-cell clones were first screened for their reactivity to IgA and cross-recognition of badger IgG, by direct ELISA against HPLC purified sIgA from badger bile and IgG purified from serum (Figure 2). Three of the fifteen supernatants (16/2, 8/1, 8/2), recognised badger IgG. All fifteen supernatants recognised sIgA purified from badger bile (Figure 2). Each monoclonal antibody (mAb) was then used in a direct ELISA against a titration of the purified sIgA. During this phase of the project, a commercial anti-ferret IgA antibody became available and was included in the direct ELISA for comparative purposes. The sensitivity of the ELISA was determined using purified badger bile titrated from 500 down to 15 ng/mL (Figure 3).

### 3.3. Recognition of Recombinant MPB83 by IgA in Serum and Tracheal Aspirate

Serum and tracheal aspirate samples from experimentally infected badgers collected 17 weeks after endobronchial infection with *M. bovis* at University College Dublin, Ireland [19] were tested against MPB83 in the MPB83-IgA ELISA (Figure 4). Animals given the highest dose of *M. bovis* (3000 cfu), exhibited the highest IgA responses by ELISA, particularly in serum (Figure 4A). All three badgers that received the highest dose of *M. bovis* were positive, being above the cut-off OD value of 0.097. Only one tracheal aspirate sample (from animal 180) was above the positive cut-off of 0.437 (Figure 4B).

### 3.4. Immunohistochemical Staining with Peroxidase Labelled IgA Monoclonal

Right bronchial LN from two badgers infected endobronchially 17 weeks earlier with *M. bovis* [20] revealed the presence of IgA producing cells. Slide A (Figure 5) from an unvaccinated, infected animal (C037) revealed a typical large granuloma with a necrotic core surrounded by a rim composed of numerous inflammatory cells, including many IgA cells. At post mortem, animal C037 had a greater distribution and severity of *M. bovis*-induced lesions compared to C067 [20]. In contrast, slide B (Figure 5) is from an infected badger previously vaccinated with BCG (C067), where small solid non-necrotic granulomas are visible with relatively few IgA positive cells.

## 4. Discussion

In humans, IgA is the predominant immunoglobulin in mucosal tissues and secretions, and the second most abundant in serum, and may be important in protection against mycobacterial infection, especially in the respiratory tract. When IgA-deficient mice were infected with BCG, there was more profound pathology in the lung of the deficient mice compared to the wild-type mice [33]. In addition, Reljic et al. demonstrated that the intranasal inoculation of mice with monoclonal IgA against the α-crystallin antigen of *M. tuberculosis* can reduce the severity of tuberculous infection in the lungs [34].

To evaluate the antimycobacterial IgA response of badgers and investigate its diagnostic potential, we produced mAbs to badger IgA and characterised them by Western blot and direct ELISA, following sIgA purification from badger bile. On the basis of size, badger sIgA appeared to be comprised of heavy and light chain components, with SC the most abundant protein. Size comparison with published data from other species showed badger HC, LC and SC to be most similar to those of baboon IgA (Table 1).

Three hybridoma supernatants recognised badger IgG, presumably through the recognition of a common epitope in the Fc portion of the immunoglobulin. All mAbs raised specifically to IgA purified from badger bile were found to be more sensitive than the commercial anti-ferret IgA mAb, despite the anticipated high similarity of ferret IgA to badger IgA. The ELISA based on the badger-specific mAbs was determined to have a detection limit of about 20 ng/mL and a linear detection range was between 15 and > 250 ng/mL. Since IgA concentrations in canine sera varied from 0.7 to 2.6 mg/mL [35], but were as low as 4 μg/mL in human bronchial secretions [36], this level of sensitivity looked promising for the development of a sensitive assay for badger IgA in secretions such as tracheal aspirate, and serum. The recognition of recombinant MPB83, an immunodominant antigen of *M. bovis* infected badgers, by IgA in serum and tracheal aspirate is a promising step towards the development of an *M. bovis*-specific assay. The limit of detection shown in the serum assay (Figure 4A) indicates that animals given the high dose of *M. bovis* and showing the most severe TB lesions [19] have the highest concentration of specific IgA. There appeared to be limited correlation between IgA levels in serum and tracheal aspirate, with only one animal positive in the latter sample type (Figure 4B). This was somewhat surprising given the respiratory nature of the infection, but might be explained by the variable amount of mucus that can be collected by aspiration. Indeed, we have separately evaluated the amount of IgA present in tracheal aspirate samples from different badgers by direct ELISA and found the amounts to be extremely variable (data not shown).

There is emerging evidence that suggests a role for B cells and humoral immunity in the control of intracellular pathogens, such as *M. tuberculosis* in humans [37]. A study conducted at APHA, reported that in BCG-vaccinated cattle, there was proliferation of B cells along the periphery and within granulomas [38]. The study of IgA and B cells in the granuloma could be a useful tool to study both the pathogenesis of disease caused by *M. bovis* and to test for a correlation between IgA levels and the severity of pathology or the protection in vaccine efficacy trials.

The most sensitive of the mAbs generated in this study (1/1 and 2/1 specific for IgA, and 8/2 and 16/2 specific for both IgA and IgG), permit the detection of antigen-specific IgA in badgers infected with *M. bovis*. Further work with these reagents is warranted to confirm they are truly specific for IgA, although the data presented here point to this being the case. The serum IgA response to bTB infection is rarely studied, although the combination of IgG and IgA detection provided the best sensitivity of TB detection in people, compared to the use of IgA and IgG alone [39]. The three mAbs with reactivity to both IgG and IgA may permit the simultaneous detection of both immunoglobulins in a test of bTB infection in badgers, whilst inclusion of *M. bovis* specific antigen, MPB83 indicates scope for development of a specific ELISA.

## 5. Conclusions

Secretory IgA purified from badger bile was used to generate badger-specific anti-IgA mAbs. The most promising mAbs selected based on titration against badger sIgA were used for the development of an *M. bovis*-specific assay based on recognition of rMPB83. By ELISA, IgA with specificity to MPB83 was detected in the serum and tracheal aspirate of experimentally-infected badgers. Peroxidase-labelled IgA was also used for the immunohistochemical detection of IgA-positive cells surrounding the necrotic core of granulomas from bTB-infected badgers. With further characterization, these represent new reagents for the study of the IgA response in badgers.

## Figures and Tables

**Figure 1 vetsci-06-00089-f001:**
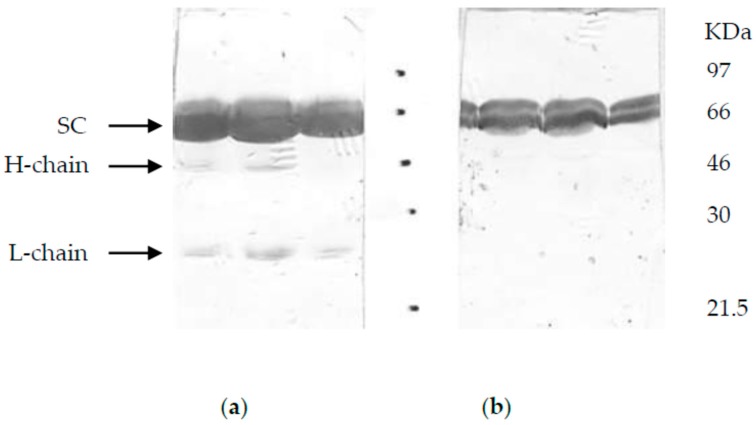
Western blot of badger sIgA. Badger sIgA was purified from bile by gel filtration and (**a**) probed using rabbit polyclonal antibodies to dog IgA (Bethyl Laboratories, UK) and (**b**) pan-species SC (Bethyl Laboratories, UK). Heavy (H) and light (L) chains of IgA indicated based on published sizes for other species.

**Figure 2 vetsci-06-00089-f002:**
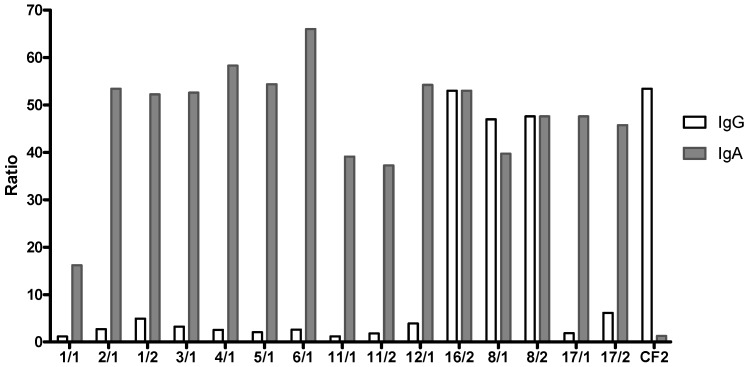
Detection of badger IgA and IgG by mAbs. Ratio value (OD value divided by the corresponding baseline value) for each mAb, showing recognition of sIgA from bile and additionally, of serum IgG by mAbs 16/2, 8/1 and 8/2. CF2 = mAb specific for badger IgG [12].

**Figure 3 vetsci-06-00089-f003:**
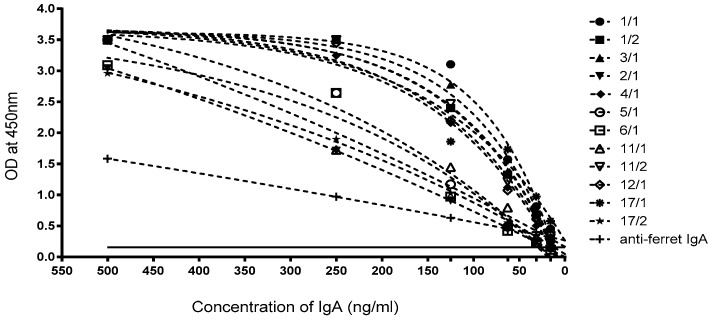
Titration of badger sIgA and recognition by anti-badger IgA monoclonals. Dashed lines show the fits to the data as one phase exponential decay curves (GraphPad Prism version 6.04 for Windows, GraphPad Software, La Jolla, CA, USA, www.graphpad.com). The goodness of fit (R^2^) for these curves varied from 0.9482 (17/1) to 0.9999 (2/1). The solid line represents the background OD, (without the purified IgA), plus twice the standard deviation.

**Figure 4 vetsci-06-00089-f004:**
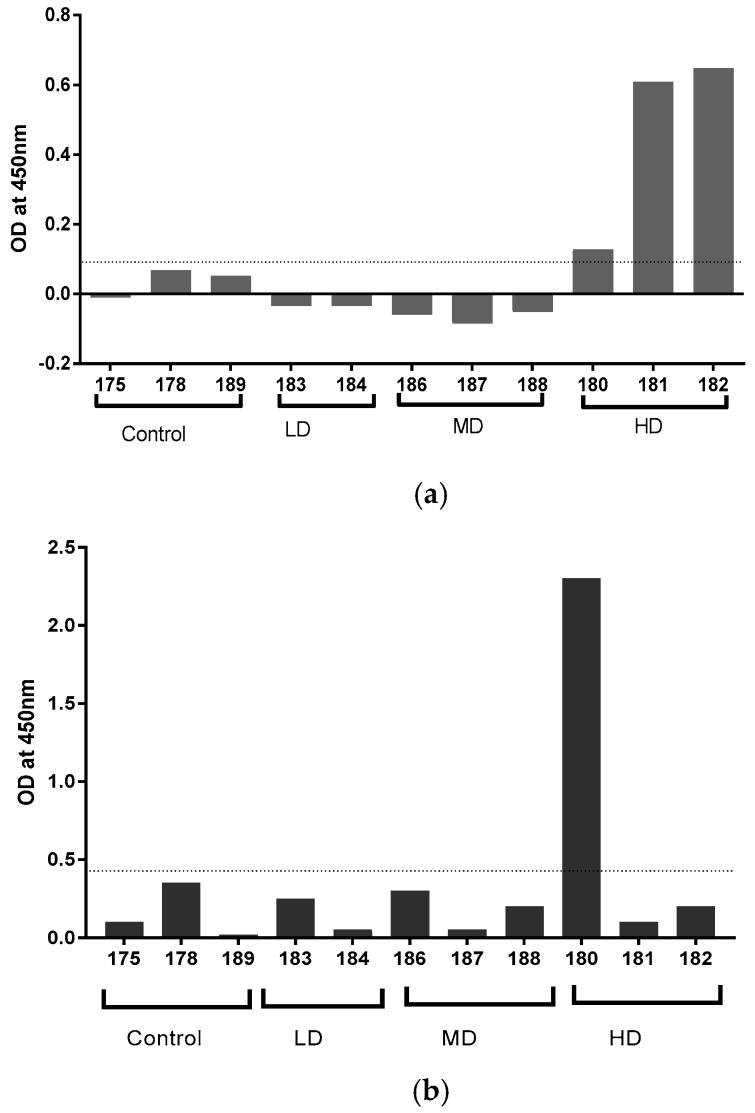
Recognition of MPB83 antigen by IgA. (**a**) Serum and (**b**) tracheal aspirate samples from badgers infected endobronchially with *M. bovis* 17 weeks previously at doses of < 10 cfu (LD), approximately 100 cfu (MD), or approximately 3000 cfu, (HD) from [19]. The dashed lines represent the mean ODs for the controls, plus twice the standard deviation.

**Figure 5 vetsci-06-00089-f005:**
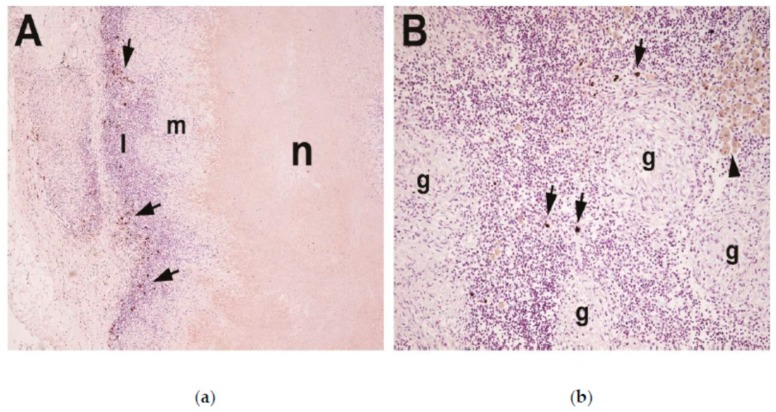
Immunohistochemical detection of IgA in TB-infected badger lymph node tissue. Tissue sections were prepared from badgers [20]. (**a**) Unvaccinated control animal (C037) with typical, large necrotic granuloma. IgA-positive cells (arrows) are observed within the outer layers of the granuloma surrounding the necrotic (n) core and the macrophage (m) layer. The majority of IgA positive cells are observed within the rim of lymphocytes (l), at 100× magnification. (**b**) Animal vaccinated with BCG (C067) with small, solid, non-necrotic granuloma. Few IgA positive cells (arrows) are observed outside small, solid, non-necrotic granulomas (g). Silica-laden macrophages (arrowhead) are also observed, at 200× magnification.

**Table 1 vetsci-06-00089-t001:** Estimated molecular weights of badger sIgA components compared with those of selected other species for which data could be found.

Species.	Source	H-Chain (kDa)	L-Chain (kDa)	SC (kDa)	Reference
Badger	Bile	46	27	66	This work
Rat	Bile Milk	50–58 NK ^1^	25–28 NK	66–80 66–71	[22]
Asian elephant	Milk	55–60	NK	68–82	[23,24]
Guinea pig	Milk	52	24	72–88	[25,26]
Human	Milk Colostrum	57–64	25–28	73–90 76	[22,27,28]
Pig	Milk	55–57	23	72–90	[29,30]
W.I. manatee	Milk	55–57	NK	70–86	[24]
Baboon	Colostrum	45–66	30	66–97	[31]
Rabbit	Colostrum	62–67	23	64	[26,32]

^1^ NK = not known.

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
