# Peer review of "Purification and Characterisation of Badger IgA and Its Detection in the Context of Tuberculosis"

_vetsci, 2019, doi:10.3390/vetsci6040089_

Round 1
Reviewer 1 Report
The European badger is an important wild animal reservoir of bovine tuberculosis in the UK. The ability to detect infected badgers might be enhanced by expanding the current repertoire of immunological assay that have been developed for that purpose. The authors choice of secretory IgA (sIgA) is logical given the likely respiratory nature of mycobacterial infection in wild badgers and the accessibility of mucosal samples in living animals.
In this manuscript, the authors claim to have purified sIgA from badger bile and generated monoclonal antibodies against that molecule in mice. They examined serum and tracheal aspirates from infected badgers for the presence of sIgA specific for a common mycobacterial protein (MPB83).
The methods utilized by the authors to purify sIgA from bile were straightforward and appropriate. However, the assays used to demonstrate that the molecule was sIgA were less than satisfactory. A skeptic would have a difficult time accepting the authors’ assertions. The following questions and suggestions might assist the authors to strengthen their supporting data.
The only portion of the IgA molecule that contains an isotype-specific epitope is the Fc portion. Why didn’t the authors cleave the whole molecules enzymatically and purify the Fc to immunize the mice? The very wide range of sizes for the H chain would include virtually all immunoglobulin isotypes. That fact, plus the very faint bands for H chains in Figure 1, do not convince the skeptical reader that the molecule is IgA. There is plenty of SC present, but that does not prove that the associated immunoglobulin is IgA. Is sIgA a dimer in badgers like it is in humans and some other species? If so, then the definitive proof that sIgA has been isolated is a non-denaturing gel (and accompanying Western blot) showing a molecule of appropriate size for a dimer associated with SC. Where is that gel and that blot? There is likely to be plenty of IgA in badger serum. Why didn’t the authors test each of the monoclonal antibodies in Figure 2 against serum as well as bile samples? Another assay to convince the skeptical reader would be to absorb the bile and serum samples with Protein G Sepharose and then show that their monoclonal antibody still detected the putative residual IgA. The authors have not provided the time post-infection at which the samples used in Figures 4 & 5 were taken. This question gets at the sensitivity (i.e., earliest detectable antibody following infection) of the assay. Perhaps the authors could obtain serial serum (and aspirate?) samples at frequent intervals post-exposure. The disparity between the serum and aspirate samples from the same animals (Figure 4) is disappointing. Did the authors demonstrate that the aspirate samples had any IgA at all, regardless of antigenic specificity?Author Response
We thank this reviewer for their constructive questions and suggestions in strengthening our supporting data.
The only portion of the IgA molecule that contains an isotype-specific epitope is the Fc portion. Why didn’t the authors cleave the whole molecules enzymatically and purify the Fc to immunize the mice? The very wide range of sizes for the H chain would include virtually all immunoglobulin isotypes. That fact, plus the very faint bands for H chains in Figure 1, do not convince the skeptical reader that the molecule is IgA. There is plenty of SC present, but that does not prove that the associated immunoglobulin is IgA. Is sIgA a dimer in badgers like it is in humans and some other species? If so, then the definitive proof that sIgA has been isolated is a non-denaturing gel (and accompanying Western blot) showing a molecule of appropriate size for a dimer associated with SC. Where is that gel and that blot? There is likely to be plenty of IgA in badger serum. Why didn’t the authors test each of the monoclonal antibodies in Figure 2 against serum as well as bile samples? Another assay to convince the skeptical reader would be to absorb the bile and serum samples with Protein G Sepharose and then show that their monoclonal antibody still detected the putative residual IgA.We agree with the reviewer, that in hindsight the approaches they suggest would have strengthened our ability to conclude that we had definitely isolated badger IgA from bile and raised monoclonal antibodies specific to IgA not other classes of immunoglobulin. Unfortunately we are not in a position to repeat this work. However, we remain confident that on the balance of probability we have indeed achieved our aims, for the following reasons: (1) the western blot shown in the first panel of Figure 1 was generated using a monoclonal with specificity to dog IgA and the three components one would expect for IgA (SC, HC, LC) all appear to be detected; (2) IgG isolated by Protein G Sepharose column was detected by our IgG-specific monoclonal CF2, which failed to detect any IgG in the material we had purified from badger bile, thus demonstrating IgG is not a detectable contaminant within the material we had isolated from badger bile (Figure 2); (3) conversely, the vast majority of monoclonals we raised to what we believed to be IgA did not detect IgG (Figure 2). Thus the presence of large amounts of SC within the material we used to raise these monoclonals strongly suggests we have generated IgA-specific reagents. The results shown in Figure 3 also suggest that monoclonal 1/1 detects the Fc portion of IgA rather than SC. We say this because 1/1 was more successful in detecting IgA in serum (where SC should be absent) than in detecting sIgA in tracheal aspirate.
The authors have not provided the time post-infection at which the samples used in Figures 4 & 5 were taken. This question gets at the sensitivity (i.e., earliest detectable antibody following infection) of the assay. Perhaps the authors could obtain serial serum (and aspirate?) samples at frequent intervals post-exposure. The disparity between the serum and aspirate samples from the same animals (Figure 4) is disappointing. Did the authors demonstrate that the aspirate samples had any IgA at all, regardless of antigenic specificity?The samples used in Figures 4 and 5 were obtained 17 weeks post-infection and this information is now included in the Results section accordingly. Apologies for this oversight.
We have separately evaluated the amount of IgA present in tracheal aspirate samples from different badgers by direct ELISA and found the amounts to be extremely variable. We now say this in lines 266-268 of the Discussion as the possible reason for poor correlation between serum and tracheal aspirate shown in Figure 4.
We do have both serum and TA samples at different time points during the course of M. bovis infection so in future work we intend to evaluate these in both our IgG and IgA ELISAs. We do have some preliminary data from wild badgers that demonstrate some badgers with TB (confirmed by M. bovis culture) are positive in the serum IgA ELISA to MPB83 whereas they are not positive in our IgG ELISA. However, this work needs to be expanded upon before being suitable for publication.
We note this reviewer indicates our conclusions need to be tempered by limitations they see in our Results. Although we hope we have addressed these concerns in this reply we have modified the text at various points in the paper to concede that further evaluation is warranted before concluding definitively that these reagents detect IgA. This includes modifications at lines 20-21, 27, 247-248, 277-278, and 290-291.Reviewer 2 Report
The manuscript described analysis of bovine tuberculosis in European badgers. One monoclonal antibody specific for badger IgA was used to detect IgA induce by antigen of M. bovis. Thus, the authors’ method will be expected to be promising analysis for bovine tuberculosis in European badgers. Therefore, the manuscript is not too excellent to be published, after schemes are completed. In other words, the manuscript is so excellent that it should be published.
Comments
(1) Do not germs different from M. bovis induce IgA in European badgers? If so, is IgA detection really correct in analysis of bovine tuberculosis?
(2) In this case, is the ELISA for badgers cheap?
(3) How much % of European badgers are suffering from bovine tuberculosis in Great Britain?
(4) In Table 1, estimated molecular weights of each animal species have some range of values. Does this mean that some of IgAs were modified by sugars and other materials? If so, is IgA detection really correct in analysis of bovine tuberculosis?
That is all.
Author Response
Comments
1. Do not germs different from M. bovis induce IgA in European badgers? If so, is IgA detection really correct in analysis of bovine tuberculosis?
Almost certainly other infections will generate IgA responses in badgers, however, these will be specific to the infection in question. We provide specificity in our assay by evaluating IgA responses to MPB83 antigen. Whilst this antigen is shared by other mycobacteria, it has been shown in numerous publications to be a specific antigen for the serodetection of M. bovis infection in badgers and other species. Figure 4 illustrates that no serum IgA responses were seen in control badgers uninfected with M. bovis, and indeed, only responses were seen to MPB83 in badgers that received the highest dose of challenge with M. bovis. Further evaluation of the specificity of IgA responses to MPB83 will require evaluation on a larger panel of sera obtained from wild badgers.
2. In this case, is the ELISA for badgers cheap?
We have not calculated the cost of the ELISA but it would not be any more expensive than the existing ELISA test based on the serodetection of IgG responses to MPB83. There is even the possibility of adding the serodetection of IgA to the IgG assay by labelling the two monoclonal antibodies with different fluorescent tags. We have some preliminary data to show that badgers with TB sometimes display an IgA response to MPB83 in the absence of an IgG response. Thereby, combining both reagents in a single ELISA we hope to be able to improve the sensitivity of the test in the most cost-effective manner.
3. How much % of European badgers are suffering from bovine tuberculosis in Great Britain?
Only estimates are possible, and the prevalence varies greatly from one geographical region to another in the UK, and even from one badger colony to another in the same area. This recent paper placed the prevalence at 21% in road traffic-killed badgers on the edge of the British bovine TB epidemic area (Sandoval Barron E, et al. Sci Rep. 2018;8(1):17206. doi: 10.1038/s41598-018-35652-5).
4. In Table 1, estimated molecular weights of each animal species have some range of values. Does this mean that some of IgAs were modified by sugars and other materials? If so, is IgA detection really correct in analysis of bovine tuberculosis?
Papers vary in the size estimates they give for IgA. This is often the consequence of differing methods used in the assays. For example, some studies use density gradient ultracentrifugation whilst others estimate size based on migration in PAGE against size markers, as we have done. Undoubtedly some of the size differences are due to post-translational modifications, such as glycosylation. The extent to which these modifications may occur in badgers and the impact that may have on test performance is not known. Interestingly, two of our monoclonals raised to badger IgA also recognized badger IgG (16/2 and 8/1) demonstrating these monoclonals were binding to a region common to both classes immunoglobulin. However, this was the minority of monoclonals. The monoclonal used for our ELISA (1/1) did not recognize badger IgG.
Round 2
Reviewer 1 Report
No additional concerns